# The Isolation and Characterization of Antagonist *Trichoderma* spp. from the Soil of Abha, Saudi Arabia

**DOI:** 10.3390/molecules27082525

**Published:** 2022-04-14

**Authors:** Aisha Saleh Alwadai, Kahkashan Perveen, Mona Alwahaibi

**Affiliations:** Department of Botany & Microbiology, College of Science, King Saud University, Riyadh 11495, Saudi Arabia; kperveen@ksu.edu.sa (K.P.); malwhibi@ksu.edu.sa (M.A.)

**Keywords:** Abha, antagonistic activity, fungi, Saudi Arabia, soil, *Trichoderma* species

## Abstract

**Background**: The genus *Trichoderma* is widely spread in the environment, mainly in soils. Trichoderma are filamentous fungi and are used in a wide range of fields to manage plant patho-genic fungi. They have proven to be effective biocontrol agents due to their high reproducibility, adaptability, efficient nutrient mobilization, ability to colonize the rhizosphere, significant inhibitory effects against phytopathogenic fungi, and efficacy in promoting plant growth. In the present study, the antagonist Trichoderma isolates were characterized from the soil of Abha region, Saudi Arabia. **Methodology:** Soil samples were collected from six locations of Abha, Saudi Arabia to isolate *Trichoderma* having the antagonistic potential against plant pathogenic fungi. The soil dilution plate method was used to isolate *Trichoderma* (*Trichoderma* Specific Medium (TSM)). Isolated *Trichoderma* were evaluated for their antagonistic potential against *Fusarium oxysporum*, *Alternaria alternata* and *Helminthosporium rostratum*. The antagonist activity was assessed by dual culture assay, and the effect of volatile metabolites and culture filtrate of *Trichoderma.* In addition, the effect of different temperature and salt concentrations on the growth of *Trichoderma* isolates were also evaluated. **Results:** The most potent *Trichoderma* species were identified by using ITS4 and ITS 5 primers. Total 48 *Trichoderma* isolates were isolated on (TSM) from the soil samples out of those six isolates were found to have antagonist potential against the tested plant pathogenic fungi. In general, *Trichoderma* strains A (1) 2.1 T, A (3) 3.1 T and A (6) 2.2 T were found to be highly effective in reducing the growth of tested plant pathogenic fungi. *Trichoderma* A (1) 2.1 T was highly effective against *F. oxysporum* (82%), whereas *Trichoderma* A (6) 2.2 T prevented the maximal growth of *H. rostratum* (77%) according to the dual culture data. Furthermore, *Trichoderma* A (1) 2.1 T volatile metabolites hindered *F. oxysporum* growth. The volatile metabolite of *Trichoderma* A (6) 2.2 T, on the other hand, had the strongest activity against *A. alternata* (45%). The *Trichoderma* A (1) 2.1 T culture filtrate was proven to be effective in suppressing the growth of *H. rostratum* (47%). The temperature range of 26 °C to 30 °C was observed to be optimum for *Trichoderma* growth. *Trichoderma* isolates grew well at salt concentrations (NaCl) of 2%, and with the increasing salt concentration the growth of isolates decreased. The molecular analysis of potent fungi by ITS4 and ITS5 primers confirmed that the *Trichoderma* isolates A (1) 2.1 T, A (3) 3.1 and A (6) 2.2 T were *T. harzianum*, *T. brevicompactum*, and *T. velutinum*, respectively. **Conclusions:** The study concludes that the soil of the Abha region contains a large population of diverse fungi including Trichoderma, which can be explored further to be used as biocontrol agents.

## 1. Introduction

Microorganisms in the soil effect the ecosystem by contributing to soil structure, soil fertility, plant health, and plant nutrition [1]. It has been widely documented in the last two decades that a larger part of the harsh environment is inhabited by varied microbial communities. Soil microorganisms include three major groups such as: bacteria, archaea and fungi. One of the most important ascomycete funguses that is found in soil is *Trichoderma*, which has the ability to produce a wide range of antibiotic substances and mycoparasite activities such as gliovirin, gliotoxin, viridin, pyrones, peptaibols, and α-aminoisobutyric acid [2]. *Trichoderma* species compete with other soil microorganisms for nutrients and space because they grow rapidly [1]. Due to the rapid growth of *Trichoderma* species and the abundance of their conidia, its isolation could be possible from soil by all available conventional methods. It has been detected to be related with plants such as epiphytes and endophytes as well as in the rhizospheric region and it plays considerable roles in plant growth promotion and soil health. Several species of *Trichoderma* have been detected up to now but many are yet to be explored [3].

*Trichoderma* species are filamentous fungi used in a wide range of fields to improve plant health for their multiple benefits, such as giving their hosts improved growth, increased disease resistance and the ability to tolerate abiotic stress [4]. Several species of *Trichoderma* (*T. virens*, *T. atroviride*, and *T. asperellum*) are capable of producing the phytohormone indole-3-acetic acid (IAA) auxin, and are used to promote plant root growth, it has been suggested [5].

It is distinguished for making signal molecules that effect the growth of plants and other fungi. Mycoparasitism is the most important mechanism against pathogens for *Trichoderma* spp. [6]. According to soil modification using this species, favorable results are shown for controlling soil-borne pathogens such as *Pythium* spp., *Fusarium* spp. and *Phytophthora* spp. and differences in antagonistic activity against the pathogen lead to inconsistencies in disease control [7]. *Trichoderma* strains have been proven to be effective biocontrol agents due to their high reproducibility, adaptability, efficient nutrient mobilization, their ability to colonize the rhizosphere, significant inhibitory effects against phytopathogenic fungi, and their efficacy in promoting plant growth. *Trichoderma* strains have been created and utilized as biocontrol agents against plant fungal infections. The strains *T. harzianum*, *T. viride*, and *T. virens* have been most commonly utilized for biological control. According to some studies, they are efficient in reducing root rots/wilt complexes and foliar diseases in a variety of crops, as well as in suppressing a variety of phytopathogens such as *Rhizoctonia*, *Pythium*, *Sclerotinia*, *Sclerotium*, *Fusarium*, and *Macrophomina* [7]. The current study was conducted to isolate and characterize antagonist *Trichoderma* sp. from the soil of Abha region, Saudi Arabia.

## 2. Results

### Isolation of Trichoderma *spp.* from the Soil of Abha, Saudi Arabia

For isolation of antagonist *Trichoderma* spp., 6 soil samples were collected from 6 different locations in Abha, Saudi Arabia. The dilution plate method was used for the isolation of fungi from soil samples. Table 1 shows the colony-forming units (CFU)/g of fungi on Potato Dextrose Agar (PDA) and *Trichoderma* Specific Medium (TSM). The fungal population isolated from different soils on the PDA plates ranged from 45.3 × 10^2^ CFU/g to 16 × 10^2^ CFU/g. The maximum number of colonies of fungi on PDA plates (45.3 × 10^2^ CFU/g) were counted from Al Areen (A3). The fungal population on the TSM plates from different soil samples ranged between 83 × 10^2^ CFU/g to 8.3 × 10^2^ CFU/g. The highest number of colonies on TSM (83 × 10^2^ CFU/g) was recorded from the soil sample collected from Ain al-Dhibah (A5).

Forty-eight isolates of *Trichoderma* were picked from TSM plates, which represent soil samples of different regions of Abha. The *Trichoderma* isolates were purified and identified microscopically. The radial growth and growth rate of these isolated *Trichoderma* spp. were determined on PDA (Table 2). The total radial growth and growth rate of these isolated *Trichoderma* varies a lot. The total radial growth of isolated *Trichoderma* isolates ranges between 9.2 cm to 84 cm, while the growth rate was between 1.4 cm/day to 11.9 cm/day at 28 °C on PDA plates.

The *Trichoderma* isolates with the most distinct cultural and microscopic traits were selected for further evaluation. A total of six *Trichoderma* isolates were analyzed further.

The results of the dual culture assay of the isolated *Trichoderma* spp. against *A. alternata, F. oxysporum* and *H. rostratum* are presented in Table 3. It shows that all isolates of *Trichoderma* were able to inhibit the growth of tested plant pathogenic fungi. The percent reduction in *A. alternata* growth by different *Trichoderma* isolates ranged between 66% and 16%. The maximum reduction in the growth of *A. alternata* was achieved by *Trichoderma* (A (1) 2.1 T). Similarly, *Trichoderma* (A (1) 2.1 T) was able to reduce maximum growth of *F. oxysporum* (82%). While in the case of the plant pathogenic fungus, *H. rostratum*, the *Trichoderma* isolate (A (6) 2.2 T) was rendered to maximum growth inhibition of *H. rostratum* (77 %).

The data of the effect of volatile metabolites of *Trichoderma* isolates on the pathogenic fungi are given in Figure 1. The highest growth inhibition of *A. alternata* by *Trichoderma* was 45% by the *Trichoderma* isolate A (6) 2.2 T, while the lowest rate of growth inhibition of *A. alternata* was 5% by the *Trichoderma* isolate A (1) 2.3 T. As for *F. oxysporum*, the highest percentage of growth inhibition by *Trichoderma* was 77% by the isolate A (1) 2.1 T, and the lowest percentage of growth inhibition of *F. oxysporum* by *Trichoderma* was 1% by the isolate A (1) 2.3 T. The rate of inhibition of the growth of *H. rostratum* by *Trichoderma* was the highest, 22% by the isolate A (1) 2.1 T, while the lowest rate of growth inhibition of *H. rostratum* was 6% by the *Trichoderma* isolate A (3) 3.1 T. (Figure 1).

The result on the effect of culture filtrate of *Trichoderma* spp. on the plant pathogenic fungi reveals that most of the *Trichoderma* isolates were able to inhibit the growth of the pathogens. The highest inhibition in the growth of *A. alternata* (29%) was recorded by the culture filtrate of *Trichoderma* (A (1) 2.3 T).

As for the *F. oxysporum*, the highest percentage of growth inhibition (16%) was achieved by the *Trichoderma* (A (3) 3.1 T). The culture filtrate of *Trichoderma* (A (1) 2.1 T) inhibited the maximum growth of *H. rostratum* (41%) (Figure 2).

The data on the effect of different temperatures on the radial growth of *Trichoderma* isolates shows that, in general, the optimum temperature for their maximum growth ranges between 26 °C and 30 °C. At temperatures of 45 °C and 50 °C, the rate of radial growth was weak. The lowest growth of *Trichoderma* was at temperatures of 50 and 45, with a mean of 5 mm for all samples (Table 4).

The data presented in Table 5 show that *Trichoderma* isolates were able to grow at all the salt concentrations tested (0%, 2%, 4%, 6%, 8%, and 10% NaCl). The radial growth was best at 2%, 4%, and 6% salinity concentrations. However, at concentrations of 8% and 10%, the radial growth was medium to weak. The lowest growth of *Trichoderma* isolates was at a salinity concentration of 10% (5 mm). The highest growth rate of *Trichoderma* isolates was 2% (85 mm) (Table 5).

The molecular identification of the most potential *Trichoderma* isolates (A (1) 2.1 T), (A (3) 3.1 T) and (A (6) 2.2 T) by ITS 4 and ITS 5 showed that they were *Trichoderma harzianum, Trichoderma brevicompactum* and *Trichoderma velutinum*, respectively. The analysis of 18S rDNA of the *Trichoderma* isolates, by amplifying internal transcribed spacer region (ITS) with primers ITS4 and ITS5 and the BlastN search revealed 100% identity with 0.0 EV to *T. harzianum* (MF871551.1) (Figure 3) *T.*
*brevicompactum* (KR094463.1) (Figure 4) and *T. velutinum* (EU280080.1) (Figure 5) in NCBI GenBank respectively.

## 3. Discussion

Fungi from the genus *Trichoderma* have been widely used in agriculture as biocontrol agents because of their mycoparasitic potential and their ability to increase plant health and protection against phytopathogens, making it a great plant symbiont. *Trichoderma’s* processes include the secretion of effector molecules and secondary metabolites, which mediate *Trichoderma’s* positive interaction with plants and provide resistance to biotic and abiotic [4].

The findings revealed that there was a good population of fungi in Abha’s soil. From this area, many *Trichoderma* were isolated such as *T. velutinum*, *T. brevicompactum* and *T. harzianum*, each with a different cultural, microscopic and radial growth pattern. Recently, researchers reported that the fungi population was between 4.19 and 4.67 CFU/g from the Riyadh region, Saudi Arabia. They also reported the presence of the genus *Trichoderma* in the region [8,9,10,11,12]. *Trichoderma* spp. can adapt to various habitats easily and, generally, they are fast growers compared to their counter parts, pathogenic fungi.

The dual culture interaction and volatile components results revealed that *Trichoderma* spp. caused appreciable inhibition of mycelia growth of tested plant pathogenic fungi, *F. oxysporum*, *A. alternata* and *H. rostratum*. These fungi are important pathogens and are of concern to plant growers as they cause various important diseases. Fusarium spp. causes root and stem rot, vascular wilt, and/or fruit rot in a number of economically important crop species. It is considered one of the most important genera of plant pathogenic fungi [13]. The Alternaria are responsible for many pre- and post-harvest diseases, especially to horticulture crops, causing diseases such as leaf spots, rots, and blights [14]. Helminthosporium spp. causes leaf spot diseases mainly on cereals like sorghum, maize, wheat, rice, and several other grasses [15]. Twelve *Trichoderma* isolates were tested for antagonistic capacity in confrontational assays with three plant pathogens (*Rhizoctonia solani, Pythium ultimum*, and *Alternaria solani*) on PDA at 28 °C for four days. The three pathogens’ mycelial proliferation was inhibited by all *Trichoderma* isolates. The percentage reduction in pathogen development ranged from 36.5 to 81.4 percent [10]. *Trichoderma* isolates were tested for antagonism in vitro against *Sclerotium rolfsii*, *Rhizoctonia solani*, and *Fusarium oxysporum* f. sp. *lentis* at three different temperatures: 25 °C, 30 °C, and 35 °C in a dual inoculation test after 5 days of incubation. All *Trichoderma* isolates suppressed all three pathogens to varying degrees at 30 °C, 25 °C, and 35 °C [16]. Depending on the degree of resistance of *C. acutatum* isolates, the volatile components of *Trichoderma harzianum* exhibit a varying inhibitory impact [17]. The effect of *T. harzianum* culture filtrate on spore germination was evaluated. The culture filtrate of *T. harzianum* T3 or T24 greatly inhibited the spore germination of the tested post-harvest pathogenic fungi [18]. Previous studies have demonstrated that before mycelia of fungi interact, *Trichoderma* sp. produces low quantities of extracellular exochitinases [19,20]. The diffusion of these enzymes disintegrates host cells. These cell fragments, in turn, cause the creation of more enzymes, which set off a chain reaction of physiological changes, allowing *Trichoderma* spp. to proliferate quickly and precisely [21]. *F. oxysporum, A. alternata*, and *H. rostratum*, were also inhibited by the culture filtrate of the six *Trichoderma spp*. that were investigated. None of them, however, were fungicidal. *Trichoderma* produces antibiotics such as trichodernin, trichodermol, harzianum A, and harzianolide, which are well known [22]. Some cell wall-degrading enzymes, such as chitinases and glucanases, damage cell wall integrity by breaking down polysaccharides, chitins, and glucanase [23,24,25].The efficiency of these metabolites is dependent on the kind, quality, and quantity of antibiotics, as well as on the inhibitory compounds released by antagonists [24]. Antibiosis is produced by the creation of volatile components and non-volatile antibiotics that are inhibitory against a spectrum of soil borne fungus: *Alternaria alternata*, *Botryotinia fuckeliana* and *Sclerotinia sclerotiorum*. These, as well as parasitism, are all possible means of antagonism used by *Trichoderma* spp. Synergism between distinct forms of action modes is also a natural scenario for fungal pathogen biocontrol. Many of these ideas have been thoroughly discussed in recent reviews [26].

The current findings revealed that the optimal temperature for isolated *Trichoderma* growth was 26 °C and 30 °C. Temperature is a key factor in controlling development, sporulation, saprophytic ability, and the synthesis of non-volatile metabolites involved in nutrition, competition, mycoparasitism, and extracellular enzymes that break down fungi’s cell walls. Although most *Trichoderma* strains are mesophilic, the optimal temperature for growth varies amongst isolates [27,28].

A study reported that the radial growth of twelve isolates of *Trichoderma* showed variation at all of the temperatures examined (20 °C, 25 °C, 30 °C, 35 °C, 40 °C, 45 °C) after three days of incubation. At 30 °C, all isolates showed a maximum radial growth of 70 [16]. *T. viride* thrived at temperatures ranging from 10 °C to 30 °C, with maximum growth at 25 °C [29]. In vitro and in vivo test settings revealed that *T. afroharzianum* T22 and *T. atroviride* P1 grew differently at 20 and 25 °C [30].

Salt stress is a significant abiotic element that has a significant impact on the soil microbial community, affecting crop productivity [31]. The present study had recorded that even at the highest concentration examined, the majority of the *Trichoderma* showed tolerance to NaCI (10%). The growth, on the other hand, was better at lower NaCl concentrations. The optimal NaCl content for *Trichoderma* growth was 2%. Similar results were reported earlier by researchers, when they demonstrated that the presence of sodium chloride (2%) in the medium alters the morphology of *T. harzianum* as well as its antagonistic potential [32]. Extrusion systems have been found in *Trichoderma* species to keep intracellular sodium levels below lethal levels for the cells [33]. *T. harzianum*, *T. viride*, and *T. koningii* were shown to be tolerant of a wide range of salt concentrations in soil samples taken from glacier sites in the Indian Himalaya [34]. Furthermore, [24] suggested that high salt concentrations be utilized to prevent the prevalence of fungal/bacterial contamination in salt tolerant *Trichoderma* stocks [35].

A study reported that twelve increased salt concentrations (NaCl) resulted in a reduction in the radial development of 12 *Trichoderma* isolates observed after five days of incubation at 28 ± 1 °C [16]. Another study found that as the concentration of NaCl was raised, so did the level of *Trichoderma* growth inhibition [36].

*Trichoderma* isolates with the best results were identified using ITS molecular analysis. They were identified as *T. harzianum*, *T. brevicompactum*, and *T. velutinum.* The antagonistic potential of these three *Trichoderma* has been demonstrated in several investigations [9,37,38,39,40].

## 4. Materials and Methods

### 4.1. Collection of Soil Samples and Isolation of Trichoderma *spp.* from the Soil of Abha, Saudi Arabia

#### 4.1.1. Soil Sampling

For the collection of soil samples, six regions in Abha, Saudi Arabia were selected (Table 6 and Figure 6). A total of six soil samples (500 g each) were collected from different ecological habitats for the isolation of *Trichoderma* spp.

#### 4.1.2. Isolation of Fungi

The soil dilution plate method was employed for the isolation of fungi. *Trichoderma* Selective Medium (TSM) and Potato Dextrose Agar (PDA) were used for the isolation of fungi. The plates were incubated at 28 °C for 5 days. The number of fungal colonies on the PDA and TSM plates was counted, and the total CFU/g of soil was calculated. Putative *Trichoderma* colonies on TSM plates were purified using two rounds of PDA subculture, and pure cultures were stored at 4 °C for further analysis [42].

### 4.2. Isolation and Characterization of Trichoderma *spp.* Antagonist to Plant Pathogenic Fungi

The pure cultures of fungi isolated from the TSM plates were observed for morphological and cultural characteristics of *Trichoderma* [43,44]. The micro characteristics such as conidia and philaids of the isolates were observed under the light microscopes and the genera of the isolated *Trichoderma* were confirmed [42].

*Trichoderma* isolates were evaluated for their potential to antagonize economically important plant pathogens, *Fusarium oxysporum, Alternaria alternata* and *Helminthosporium*
*rostratum*. Identified plant pathogenic fungi cultures were obtained from the Department of Botany and Microbiology, King Saud University, Riyadh, Saudi Arabia. The antagonist activity of *Trichoderma* against *F. oxysporum*, *A. alternata* and *H. rostratum* were assayed by employing a dual culture assay, an effect of volatile metabolites of *Trichoderma*, and the effect of the culture filtrate of [37,38].

### 4.3. Effect of Different Temperature on the Growth of Trichoderma Isolates

The selected *Trichoderma* isolates were inoculated on the PDA and incubated at 26 °C, 30 °C, 45 °C and 50 °C for 7 days. The growth of *Trichoderma* isolates were determined by measuring the radial colony growth on the media [31].

### 4.4. Effect of Different Salt Concentration on the Growth of Trichoderma Isolates

The selected *Trichoderma* isolates were inoculated on the PDA containing on 0%, 2%, 4%, 6%, 8% and 10% NaCl, and were incubated at 28 °C for 7 days. The growth of *Trichoderma* isolates was determined by measuring the radial colony growth on the media [31].

## 5. Molecular Identification

### 5.1. DNA Extraction

From the pure culture of each fungus, a small lump of mycelia was collected and transferred by a sterile toothpick into a 1.5 mL. Eppendorf tube containing a lysis buffer (400 mM Tris-HCL (pH 8), 60 mM EDTA-pH 8.0, 150 mM NaCl and 1% sodium dodecyl sulfate). The tube was vortexed to disrupt the mycelia and was kept for 10 min at room temperature after that 150 µL of potassium acetate followed by brief vertexing. The sample was centrifuged at >13,000× *g* for 1 min. The supernatant was transferred to another 1.5 mL Eppendorf tube and centrifuged again as described above. The supernatant fluid was transferred into a new 1.5 mL Eppendorf tube and an equal volume of isopropyl alcohol was added. The tube was thoroughly mixed by inversion and centrifuged at >13,000× *g* for 2 min and the supernatant was neglected. The eventual DNA pellet was washed in 300 µL of 70% ethanol, after which it was spun at 10,000 rpm for 1 min and the supernatant was discarded. The DNA pellet was air-dried and dissolved in 50 µL of 1xTris- EDTA. 1 µL of purified DNA was used in 24 µL of PCR mixture [44].

### 5.2. PCR Amplification

Molecular identification of *Trichoderma* spp. was carried out based on conserved ribosomal internal transcribed spacer (ITS). The ITS regions between the small nuclear 18S rDNA and large nuclear 28S rDNA, including 5.8S rDNA, were amplified. The forward (ITS-4F) and reverse (ITS-5R) primers (Table 7) used in the PCR reactions to amplify the ITS region of the rRNA operon have been described in [45].

For all PCR mixture contained 10 µL of Red taq ready mix, 0.5 µL of each primer pair, 8 µL of analytical grade sterile water (Sigma–Aldrich) and 5 µL of genomic DNA in a total volume of 24 µL. The thermocycling program used was an initial denaturation (94 °C for 5 min), 30 cycles of denaturation (94 °C for 1 min), annealing (60 °C for 1 min) and elongation (72 °C for 1 min), and then a stabilization (72 °C for 5 min) [45]. The molecular analysis was carried out by Macrogen, Korea.

## 6. Conclusions

The study concludes that the soil of the Abha region a large population of diverse fungi. Different species of Trichoderma were isolated from the soil. *T. harzianum*, *T. brevicompactum*, and *T. velutinum* showed antagonist ability against plant pathogenic fungi, *A. alternata*, *F. oxysporum* and *H. rostratum*. The volatile compounds of *Trichoderma* isolates also had the potential to reduce the growth of plant pathogenic fungi. The optimum temperature for their growth was 26 °C. They can tolerate up to 10% salt concentration with best growth at 2%. These *Trichoderma* species can be explored further to be used as biocontrol agents.

## Figures and Tables

**Figure 1 molecules-27-02525-f001:**
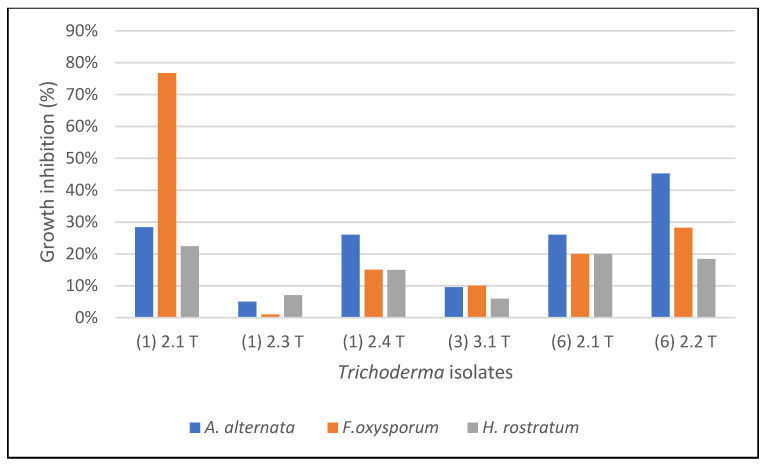
Effect of volatile metabolites of *Trichoderma* on the growth of plant pathogenic fungi.

**Figure 2 molecules-27-02525-f002:**
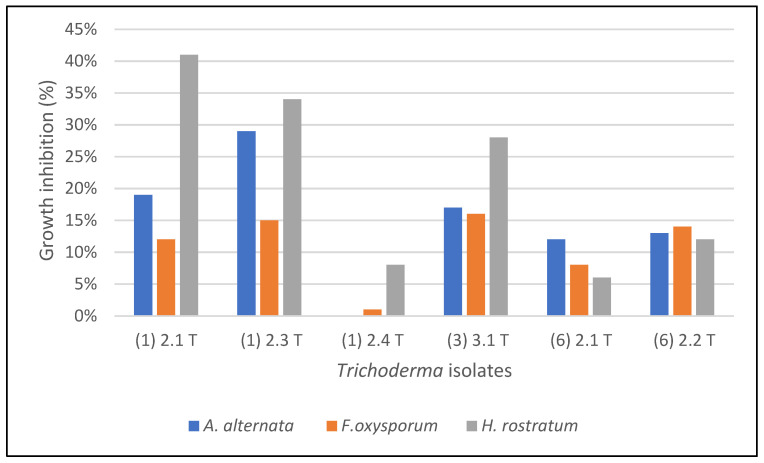
Inhibitory effect of the culture filtrate of *Trichoderma* isolates.

**Figure 3 molecules-27-02525-f003:**
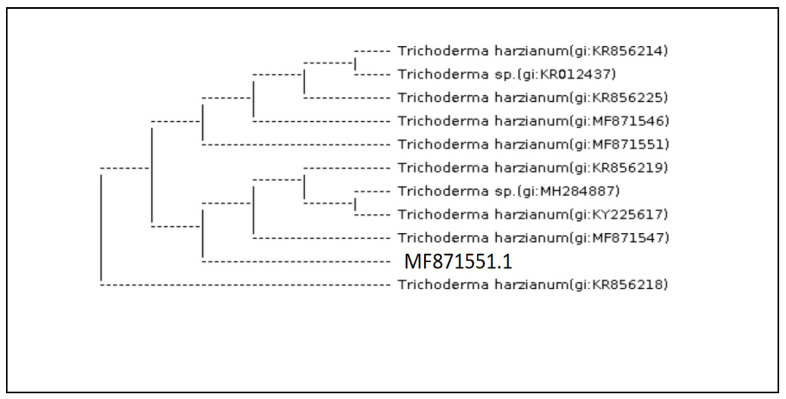
*Trichoderma* isolate (A (1) 2.1 T) (analysis of 18S rDNA of the *T. harzianum* (MF871551.1), with primers ITS4 and ITS5 in NCBI GenBank respectively).

**Figure 4 molecules-27-02525-f004:**
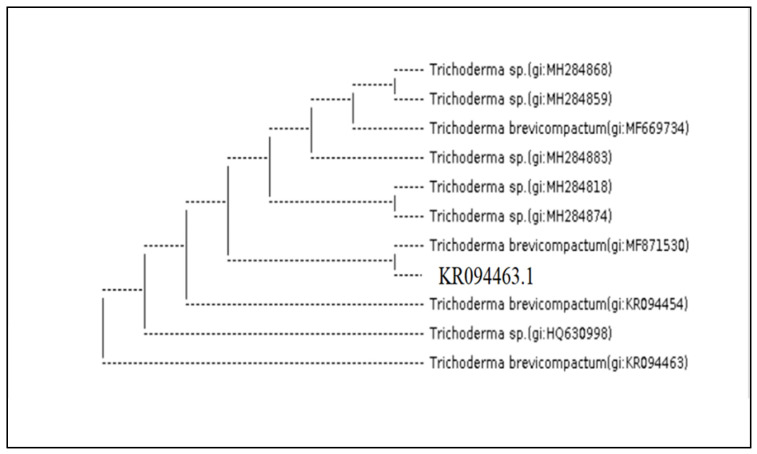
*Trichoderma* isolate (A (3) 3.1 T) (analysis of 18S rDNA of the *T. brevicompactum* (KR094463.1), with primers ITS4 and ITS5 in NCBI GenBank respectively).

**Figure 5 molecules-27-02525-f005:**
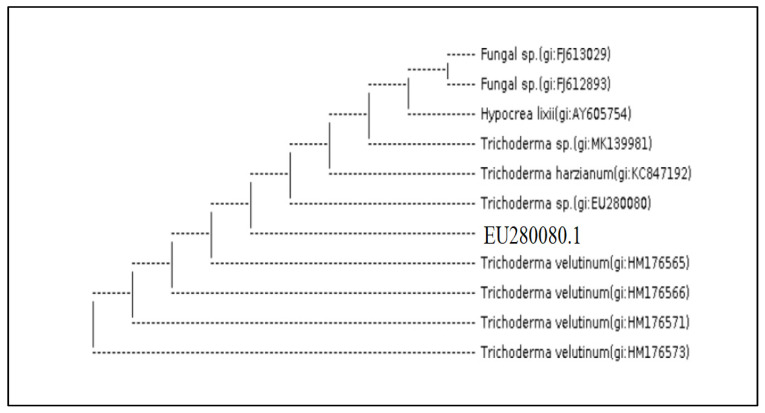
*Trichoderma* isolate (A (6) 2.2 T) (analysis of 18S rDNA of the *T. velutinum* (EU280080.1), with primers ITS4 and ITS5 in NCBI GenBank respectively).

**Figure 6 molecules-27-02525-f006:**
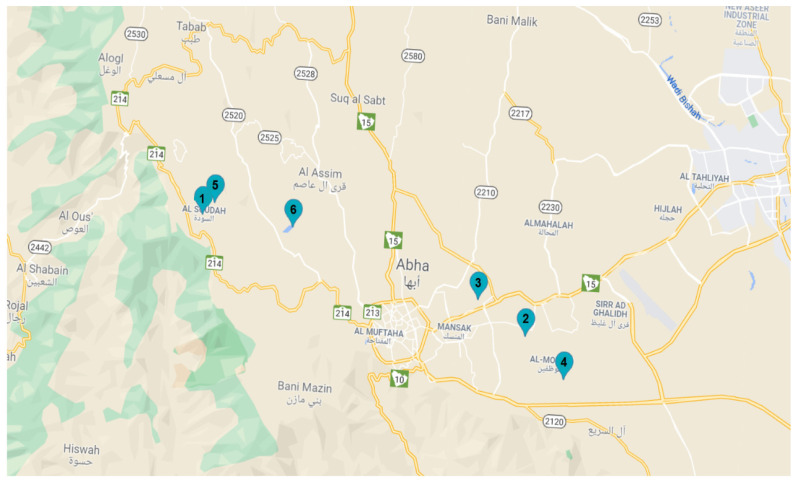
Location of soil samples collections sites, Abha, Saudi Arabia.

**Table 1 molecules-27-02525-t001:** The colony-forming units (CFU) of total fungi on PDA and *Trichoderma* on TSM in the soil samples collected from the Abha region.

S. NO	Soil Sample Code	CFU/g of Soil on PDA (10^2^)	CFU/g of Soil on TSM (10^2^)
1	A1	21	31
2	A2	16	8.3
3	A3	45.3	52
4	A4	21	16.3
5	A5	25.3	83
6	A6	27.3	56.3

**Table 2 molecules-27-02525-t002:** Total radial growth (mm) and growth rate (mm/day) of different isolates of *Trichoderma* spp., on PDA.

S. No	Isolate Code	Total Radial Growth (mm)	Growth Rate (mm/day)
1	A (1) 1.1	81.2	11.6
2	A (1) 2.2	54.6	7.8
3	A (1) 1.2 T	80.5	11.5
4	A (1) 1.4 T	79.8	11.4
5	A (1) 2.1	25.2	3.6
6	A (1) 2.1 T	85	28.7
7	A (1) 2.3 T	85	6.7
8	A (1) 2.4 T	85	14.5
9	A (1) 3.1 T	13.3	1.9
10	A (1) 3.2 T	9.8	1.4
11	A (1) 3.3 T	45.5	6.5
12	A (1) 3.4 T	44.1	6.3
13	A (2) 1.1 T	42	6
14	A (2) 1.2 T	27.3	3.9
15	A (2) 1.3 T	70.7	10.1
16	A (2) 1.4 T	39.9	5.7
17	A (2) 2.1 T	53.9	7.7
18	A (2) 3.1 T	66.5	9.5
19	A (2) 3.2 T	81.9	11.7
20	A (3) 1.1 T	14.7	2.1
21	A (3) 1.2 T	57.4	8.2
22	A (3) 1.3 T	36.4	5.2
23	A (3) 2.1 T	65.1	9.3
24	A (3) 2.2T	53.9	7.7
25	A (3) 2.3 T	37.8	5.4
26	A (3) 3.1 T	85	14.5
27	A (3) 3.2 T	52.5	7.5
28	A (3) 3.4 T	64.4	9.2
29	A (4) 1.1	81.2	11.6
30	A (4) 1.2 T	47.6	6.8
31	A (4) 2.1	56.7	8.1
32	A (4) 2.1 T	45.5	6.5
33	A (4) 2.2 T	49.7	7.1
34	A (4) 3.2	46.2	6.6
35	A (5) 1.1 T	81.9	11.7
36	A (5) 1.2 T	77.7	11.1
37	A (5) 1.4 T	72.1	10.3
38	A (5) 2.1 T	28	4
39	A (5) 2.2 T	73.5	10.5
40	A (5) 3.1 T	43.4	6.2
41	A (5) 3.2 T	50.4	7.2
42	A (5) 3.3 T	30.1	4.3
43	A (5) 3.4 T	27.3	3.9
44	A (6) 1.2 T	83.3	11.9
45	A (6) 1.3 T	84	12
46	A (6) 2.1 T	6.3	79
47	A (6) 2.2T	8.5	185
48	A (6) 3.1 T	81.2	11.6

**Table 3 molecules-27-02525-t003:** Antagonistic activity of *Trichoderma* spp. against *A. alternata*, *F. oxysporum*, and *H. rostratum* evaluated by dual culture interaction.

Code	*A. alternata*	*F. oxysporum*	*H. rostratum*
Percent Inhibition (%)			
A (1) 2.1 T	66 ± 0.82	82 ± 0.47	40 ± 0.82
A (1) 2.3 T	28 ± 0.82	19 ± 0.82	10 ± 1.25
A (1) 2.4 T	16 ± 0.82	22 ± 0.82	35 ± 0.82
A (3) 3.1 T	61 ± 0.82	64 ± 0.82	51± 0.47
A (6) 2.1 T	55 ± 1.25	19 ± 0.82	18 ± 0.82
A (6) 2.2 T	57 ± 1.25	62 ± 0.00	77 ± 0.82

**Table 4 molecules-27-02525-t004:** Effect of different temperature on the growth of *Trichoderma* isolates.

Code	Temp.	Radial Growth (mm)
A (1) 2.1 T	26 °C	85 ± 0.00
	30 °C	85 ± 0.00
	45 °C	5 ± 0.00
	50 °C	5 ± 0.00
A (1) 2.3 T	26 °C	85 ± 2.05
	30 °C	37 ± 23.37
	45 °C	5 ± 0.00
	50 °C	5 ± 0.00
A (1) 2.4 T	26 °C	85 ± 0.00
	30 °C	85 ± 2.36
	45 °C	5 ± 0.00
	50 °C	5 ± 0.00
A (3) 3.1 T	26 °C	85 ± 0.00
	30 °C	40 ± 18.71
	45 °C	5 ± 0.00
	50 °C	5 ± 0.00
A (6) 2.1 T	26 °C	79 ± 2.62
	30 °C	7 ± 0.47
	45 °C	5 ± 0.00
	50 °C	5 ± 0.00
A (6) 2.2 T	26 °C	85 ± 0.00
	30 °C	85 ± 5.73
	45 °C	5 ± 0.00
	50 °C	5 ± 0.00

**Table 5 molecules-27-02525-t005:** Effect of different salt concentrations on the growth of *Trichoderma* isolates.

Code	Salinity	Radial Growth (mm)
A (1) 2.1 T	0%	85 ± 0.00
	2%	85 ± 0.00
	4%	75.7 ± 3.30
	6%	55.3 ± 1.70
	8%	24.3 ± 1.25
	10%	8.3 ± 0.47
A (1) 2.3 T	0%	85 ± 0.00
	2%	56 ± 1.41
	4%	57.7 ± 2.05
	6%	57 ± 16.57
	8%	54.2 ± 2.36
	10%	34.3 ± 11.15
A (1) 2.4 T	0%	85 ± 0.00
	2%	53 ± 2.45
	4%	49 ± 9.90
	6%	43.7 ± 4.64
	8%	37.7 ± 12.28
	10%	25 ± 4.55
A (3) 3.1 T	0%	85 ± 0.00
	2%	74.7 ± 2.62
	4%	33.3 ± 4.03
	6%	27.3 ± 5.25
	8%	6 ± 0.00
	10%	5 ± 0.00
A (6) 2.1 T	0%	85 ± 0.00
	2%	36.3 ± 13.96
	4%	26.7 ± 3.30
	6%	43.3 ± 13.02
	8%	20.3 ± 0.47
	10%	22.7 ± 2.36
A (6) 2.2 T	0%	85 ± 0.00
	2%	85 ± 0.00
	4%	51 ± 6.38
	6%	34 ± 2.16
	8%	9.7 ± 0.94
	10%	5 ± 0.00

**Table 6 molecules-27-02525-t006:** The location of soil sample collections sites, Abha, Saudi Arabia.

S. No.	Code	Location of Sample Collection [41]	Date of Collection
1	(A1) Forest	Al-soudah	2 Junly 2019
2	(A2) Agricultural soil	Al-badiei	2 Junly 2019
3	(A3) Agricultural soil	Al Areen	2 Junly 2019
4	(A4) Rocky soil	Almuazafina	2 Junly 2019
5	(A5) Soil near the water	Ain al-Dhibah	16 Junly 2019
6	(A6) Soil near the water	Sad wadi eashran	15 Junly 2019

**Table 7 molecules-27-02525-t007:** The ITS primer pair was used in the PCR reactions to amplify the ITS region.

Primer	Sequence 5′ 3′
ITS5 (forward)	GGAAGTAAAAGTCGTAACAAGG
ITS4 (reverse)	TCCTCCGCTTATTGATATGC

## Data Availability

All the data is available in the manuscript and the soil samples were collected from Abha, Saudi Arabia.

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
