# Peer review of "The Isolation and Characterization of Antagonist Trichoderma spp. from the Soil of Abha, Saudi Arabia"

_molecules, 2022, doi:10.3390/molecules27082525_

Round 1
Reviewer 1 Report
This paper isolated several Trichoderma species, which can be used as potential biocontrol agents of fungal pathogens. Most parts of the paper are sound except for the identification of Trichoderma species. Accurate identification of Trichoderma species will help us clarify the biological properties and antagonistic activity of certain species. It is really difficult for us to identify species of the genus Trichoderma simply from ITS sequences. Thus, I have listed several related papers, and they can help you a lot when you identify Trichoderma species. More appropriate barcodes, such as rpb2 and tef1 from other species in this genus, can be found. This information may be useful for your species identification. Moreover, please check papers published by Voglmayr and his colleagues, and you can get more information on species identification in Trichoderma. Several additional comments are listed as following:
-
Abstract need to be improved, especially background information;
-
Materials and methods part, especially Part 5 (molecular identification), needs to be improved. More proper barcodes, such as rpb2 and tef1, should be used for species identification. Results parts and figures 4 to 6 must to be corrected accordingly.
Related Paper:
Biodiversity of Trichoderma (Hypocreaceae) in Southern Europe and Macaronesia
In honor of John Bissett: authoritative guidelines on molecular identification of Trichoderma
Author Response
Point 1: Abstract need to be improved, especially background information;
Response1: The abstract has been improved. More information has been included in the background section.
Point 2 : Materials and methods part, especially Part 5 ( molecular identification), needs to be improved. More proper barcodes, such as rpb2 and tef1, should be used for species identification.Results parts and figures 3 to 5 to be corrected accordingly.
Response 2 : The identified species of Trichoderma are common known species. The identity of these species was confirmed by the molecular primers ITS. According to a BLAST search, they have shown 100% similarity to the previously identified species. At present, it is not possible for the authors to conduct further molecular analysis to include suggested barcodes. We will include the recommendations of the reviwers in future studies.

Reviewer 2 Report
This manuscript "The Isolation and characterization of antagonist Trichoderma spp. from the soil of Abha, Saudi Arabia" is an interesting piece of work worth to be considering in molecules but it needs a lot of work before it is accepted. The comments and suggestions are annotated in the manuscript. Authors are advised to go through the comments carefully and address them properly.

Author Response
Point 1: This manuscript " The Isolation and characterization of antagonist Trichoderma spp. from the soil of Abha, Saudi Arabia" is an interesting piece of work worth to be considering in molecules but it needs a lot of work before it is accepted. The comments and suggestions are annotated in the manuscript. Authors are advised to go through the comments carefully and address them properly.
Response 1: All the comments and have been addressed properly in the revised manuscript. The responses to all the reviewers' comments and suggestions are included in the attached file.
